# Interobserver Reliability of Pirani and Dimeglio Scores in the Clinical Evaluation of Idiopathic Congenital Clubfoot

**DOI:** 10.3390/children8080618

**Published:** 2021-07-21

**Authors:** Vito Pavone, Andrea Vescio, Annalisa Culmone, Alessia Caldaci, Piermario La Rosa, Luciano Costarella, Gianluca Testa

**Affiliations:** Department of General Surgery and Medical Surgical Specialties, Section of Orthopaedics and Traumatology, University Hospital Policlinico “Rodolico-San Marco”, University of Catania, 95123 Catania, Italy; annalisa.culmone@libero.it (A.C.); alessia.c.92@hotmail.it (A.C.); larosapiermario@gmail.com (P.L.R.); lcostarella@yahoo.it (L.C.); gianpavel@hotmail.com (G.T.)

**Keywords:** clubfoot, Pirani score, Dimeglio score, interobserver reliability, congenital talipes equinovarus

## Abstract

Background: Dimeglio (DimS) and Pirani (PirS) scores are the most common scores used in congenital talipes equinovarus (CTEV) clinical practice. The aim of this study was to evaluate the interobserver reliability of these scores and how clinical practice can influence the clinical outcome of clubfoot through the DimS and Pirs. Methods: Fifty-four feet were assessed by six trained independent observers through the DimS and PirS: three consultants (OS), and three residents (RS) divided into three pediatric orthopaedic surgeons (PeO) and three non-pediatric orthopaedic surgeons (NPeO). Results: The PirS and DimS Scores were strongly correlated. In the same way, OS and RS, PirS, and DimS scores were strongly correlated, and the interobserver reliability ranked “good” in the comparison between PeO and NPeO. In fully trained paediatric orthopaedic surgeons, an “excellent” interobserver reliability was found but was only “good” in the NPeO cohort. Conclusions: In conclusion, after careful preparation, at least six months of observation of children with CTEV, PirS and DimS proved to be valid in terms of clinical evaluation. However, more experience with CTEV leads to a better clinical evaluation.

## 1. Introduction

Congenital talipes equinovarus (CTEV) is one of the most common congenital pediatric orthopaedic deformities, characterised by dorsal hyperflexion of the foot, varus hindfoot, adduction of the hindfoot compared to the forefoot, and increased plantar arch [1]. Clinical manifestations might depend on aetiology [1], severity, and clinical course [2,3].

To achieve uniform clinical assessments and to provide the number of casts prediction or the tenotomy need, several authors have proposed different evaluation scores. In the early 1980s, Ponseti and Smoley [4] described a clinical manifestation-based tool aimed to assess the CTEVs and provide prognosis information. Their classification system was based on ankle dorsiflexion, heel varus, forefoot supination, and tibial torsion [2]. In 1983, Harrold and Walker [5] designated a deformity correction-based classification. Subsequently, Catterall et al. described a new score [6]; the questionnaire considered four patterns depending on the evolution of the deformity. Goldner et al. [7] reported a detailed score system that grades feet from 1 to 100. The authors provided a valid tool in the recurrent deformity prediction despite the significant variation in learning curve and reliability. Moreover, 18 out of 100 points are determined radiographically. In 1995, Dimeglio et al. [8] (DimS) designed a questionnaire with the purpose of analysing the severity to obtain reference points, assess the efficacy of orthopaedic treatment, and analyse the operative results objectively. In a retrospective classification systems comparison, Wainwright et al. [2] individuated DimS as a complex and effective score with the best agreement. In the same year, Pirani et al. (PirS) [9] included most of the elements of Catterall’s method, but it is simpler to use for statistical comparisons [10].

At present, the most used evaluation scores are DimS and PirS [11,12,13], and these play a preponderant role in the clubfoot diagnosis and the treatment with the Ponseti.

The aim of our study is to evaluate the interobserver reliability of PirS and DimS scores and to evaluate how clinical practice can influence the proper severity of CTEV assessment. It was hypothesised that PirS and DimS are valid tools in clubfoot assessment with good interobserver reliability. Moreover, more skilled surgeons could report a superior observers’ agreement.

## 2. Materials and Methods

### 2.1. General Information

Between 1st September 2019 and 26th April 2020, a review of all infants younger than 6 months of age at the first cast who underwent the Ponseti method for idiopathic CTEV at our institution was carried out. All patients were admitted through the pediatric orthopaedic ambulatory with the following demographic and clinical data captured: gender, age at treatment, the involved side, and presence or absence of associated syndromes or deformities. In addition, numbers of casts and age at the surgery were collected from the medical records. The inclusion criteria were as follows: (1) confirmed diagnosis of CTEV; (2) bilateral CTEV; (3) chronological age under six months; (4) treatment with Ponseti method; and (5) complete adherence to casting program.

The exclusion criteria were neurologic and syndromic clubfeet and postural deformities; patients older than six months of age; initial treatment or previous surgery in other institutions; follow-up less than six months; and incomplete adherence to the casting program.

All cases were treated by the same paediatric orthopaedic team with experience in clubfoot treatment using the Ponseti method.

### 2.2. Clinical Severity Assessments

#### 2.2.1. Dimeglio Score (DimS)

Four clinical signs (varus, equinus, midfoot adduction, and derotation) of the calcaneo-forefoot block are evaluated in DimS. A points system based on reducibility on the relative plane, from zero to four, can be assigned to each item. Additional points were added for pre-operative deep posterior crease (1 point), deep medial crease (1 point), cavus (1 point), and muscle abnormalities (1 point). The final score can range from zero to 20 points, where a higher score indicates a more severe deformity. The severity of the deformity is then graded I–IV based on this scoring [8].

#### 2.2.2. Pirani Score (PirS)

PirS assesses six clinical signs characterising clubfoot, three items for the midfoot, and three for the hindfoot: medial crease (MC-Pir), lateral part of the head of the talus, the curvature of the lateral border, posterior crease, empty heel, and rigid equinus. Each of the six items are scored on a three-point scale (0 = none, 0.5 = moderate, 1 = severe abnormality). The total score ranges from 0 to 6 based on the severity of the deformity of the examined foot [10].

#### 2.2.3. Evaluation Contributors

All CTEV children included in the study were independently examined and assessed by three different orthopaedic surgeons (OS) and three resident doctors (RD) involved in pediatric orthopaedic care. All evaluators had previous experience of at least six months with these scoring systems. Three assessors, two OS and an RD, had a complete, full-trained program in pediatric orthopaedics (PeO) and treated more than 20 CTEV patients in the previous two years. An OS had a complete, full-trained program in foot and ankle diseases, while two RDs had at least one year of experience in the clubfoot treatment (NPeO). All the observers underwent 1 h of theoretical CTEV clinical manifestation and scoring system training before the CTEVs assessment.

### 2.3. Primary Outcome Measurement

To assess the interobserver reliability of PirS and DimS at different severities of the deformity, the intra-class correlation coefficients (ICC) statistic test was performed.

### 2.4. Secondary Outcome Measurement

To assess the importance of the experience, two cohort comparisons were performed. In the first, we compared the OS and RS; in the second, we compared the PeO and NPeO results.

### 2.5. Statistical Analysis

Continuous data are presented as means and standard deviations, as appropriate. The ICC (two-way random effects model, single-measure reliability) was performed to evaluate the observers’ agreement. According to the Koo and Li [14] guideline, agreement below 0.50 was considered as “poor”; between 0.50 and 0.74 as “moderate”; between 0.75 and 0.89 as “good”; and above 0.90 as “excellent”. The Pearson correlation coefficient (PCC) was utilised to assess the correlation between the scores. The PCC between 0.0 and 0.09 was considered “negligible”; between 0.1 and 0.39 as “weak”; between 0.40 and 0.69 as “moderate”; between 0.7 and 0.89 as “strong”, and between 0.9 and 1.0 as “very strong” [15]. A Bland and Altman plot was produced to analyse the differences between the cohorts measurements. The limits of agreement (LOA) were calculated as the mean difference ± 1.96 standard deviations (SD) [16].

## 3. Results

Twenty-seven (18 females and 9 males) patients were considered eligible and included in the study. A total of 54 feet were assessed by six independent observers. The mean age at the first cast was 22 ± 11 days. The mean PirS at the first cast was 4.9 ± 1.0, while the mean DimS was 3.2 ± 0.9. The median of cast numbers was 6 ± 2. In all cases, the Achilles tenotomy was performed.

### 3.1. PirS and DimS Interobserver Reliability

According to the Koo and Li guideline [14], the PirS ICC between the six observers was 0.80 (95% confidence interval 0.69–0.86), classified as “good”. The DimS ICC observed was 0.81 (95% confidence interval 0.74–0.87) and considered as “good”. A “strong” correlation between the scores was found according to Schober et al. classification (PCC = 0.89 (*p* < 0.001)).

### 3.2. OS and RS Scores’ Interobserver Reliability

The PirS ICC between the OS cohort was 0.80 (95% confidence interval 0.70–0.86), while RS PirS ICC was 0.82 (95% confidence interval 0.74–0.89), and both were considered “good” (Figure 1).

The DimS ICC observed was classified as “good” for both the cohorts (OS ICC = 0.78 (95% confidence interval 0.68–0.86); RS ICC = 0.80 (95% confidence interval 0.70–0.87)) (Figure 2).

A strong correlation was observed according the PirS PCC (0.87 (*p* < 0.001)) and DimS PCC was 0.85 (*p* < 0.001) and classified as “strong”.

### 3.3. PeO and NPeO Scores’ Interobserver Reliability

The PirS ICC between the PeO cohort was 0.95 (95% confidence interval, 0.92–0.97) and considered “excellent”, while NPeO PirS ICC was 0.76 (95% confidence interval, 0.66–0.84) and classified as “good” (Figure 3).

The DimS ICC observed was classified as “excellent” for PeO group (ICC = 0.91 (95% confidence interval 0.87–0.95), while NPeO DimS ICC was 0.80 (95% confidence interval 0.70–0.87) and considered “good” (Figure 4).

Considering the PCC, strong correlations were found (PirS PCC = 0.76 (*p* < 0.001); DimS PCC = 0.75 (*p* < 0.001)).

## 4. Discussion

Based on our data, PirS and DimS are valid scores in the assessment of CTEV clinical manifestation. Both scores were strongly correlated and showed good interobserver reliability. Moreover, our findings highlighted that measurement training could play a crucial role in clinical evaluation. In pediatric orthopaedic disorders assessed by experienced observers, greater concordance in the assessment was found. Few studies have attempted to evaluate PirS and DimS score reliability assessments [2,8,10,11,12,17,18,19,20,21,22,23], and the comparison between the studies is challenging due to the different statistical evaluation methods used to calculate the agreement degree among the observers. Vasu et al. [24] analysed the prognostic efficacy of both scores and reported a high validity in terms of clinical and prognostic evaluation but not the superiority of a score. Mulker et al. [25] and Mejaby [13] confirmed the high prognostic validity of DimS and Pirs, respectively. Several authors [17,19,26] have verified the high reliability of the scores. According to previous findings, our data demonstrated a strong correlation between the scores and good interobserver reliability. Interestingly, in two studies [17], the authors investigated the inter-rater reliability of the total score and of the sub-parameters. Although both trials reported a high correlation of the total score of both questionnaires, a few sub-parameters were found with a low reliability degree. In particular, according to the kappa values (acceptable if >0.60), every sub-parameter highlighted poor reliability values except for the equinus and the curvature of the lateral border in PirS, 0.74 for the muscle abnormality in DimS [26].

Other authors [20,23] have investigated the role of experience in CTEV assessment but evaluated only a single questionnaire, the PirS. Shaheen et al. [20] analysed the interobserver reliability of PirS between a pediatric orthopaedic surgeon and a physiotherapy assistant and observed a moderate to substantial concordance [20]. Sharma et al. [23] enrolled orthopaedic surgeons, a resident doctor, and a nonmedical counsellor. The authors highlighted discreet-to-remarkable interobserver reliability of all the subcomponents. In our study, three orthopaedic surgeons and three orthopaedic residents were compared. In both cohorts, the interobserver reliability was “good”. Moreover, a strong correlation was observed according to the PirS and DimS among the groups.

To our knowledge, this is the first study comparing pediatric orthopaedics trained surgeons and general orthopaedic surgeons. Despite good reliability and a strong correlation in the NPeO group, a remarkable degree of agreement was observed when comparing a full pediatric orthopaedic training and when more than 20 clubfeet were treated. It can be assumed the routine practice in clubfoot treatment can provide a more unbiased assessment than the simple training lessons.

The main limit of the study was the lack of individual sub-parameters evaluation. The study presents several strengths, including the number of observers.

## 5. Conclusions

In conclusion, PirS and DimS are valid scores for the clinical and prognostic evaluation of CTEV and have shown high interobserver reliability. After sufficient training, both scores are easily utilised in the CTEV clinical evaluation even in the less expert subject. On the other hand, orthopaedics with more practice in the treatment of clubfoot evidenced a superior concordance.

## Figures and Tables

**Figure 1 children-08-00618-f001:**
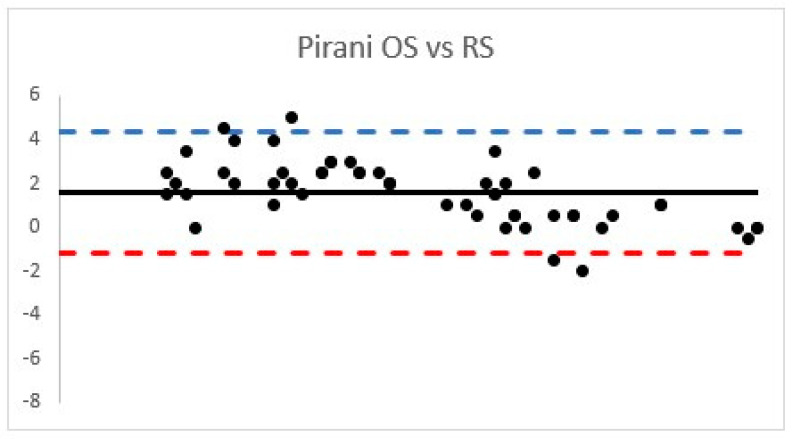
Bland Altman plots according to OS and RS Pirani score.

**Figure 2 children-08-00618-f002:**
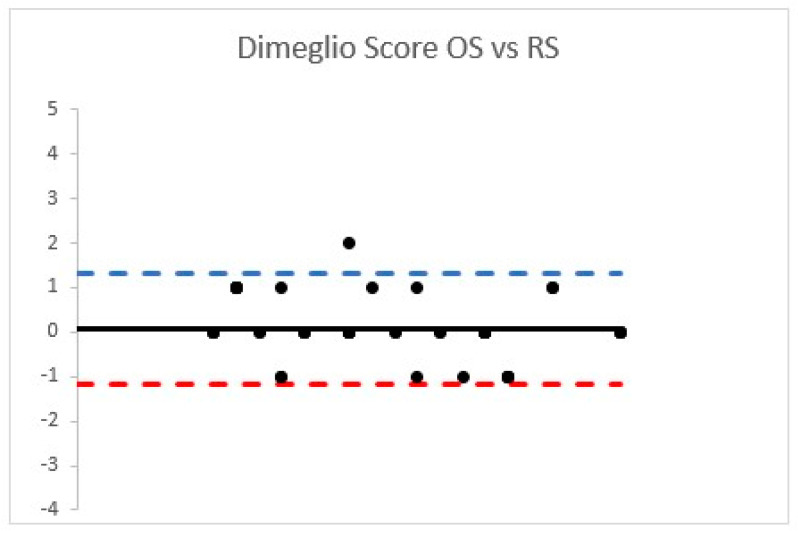
Bland Altman plots according to OS and RS Dimeglio score.

**Figure 3 children-08-00618-f003:**
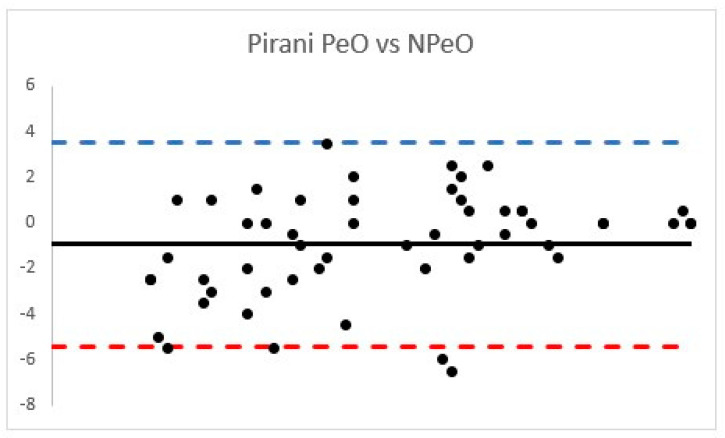
Bland Atman plots according to OS and RS Pirani score.

**Figure 4 children-08-00618-f004:**
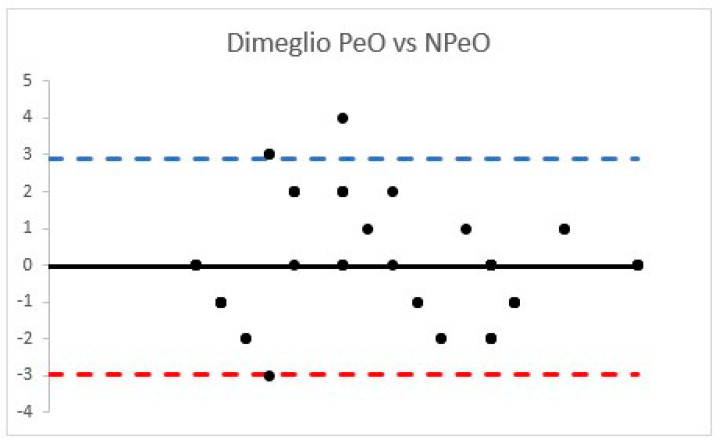
Bland Altman plots according to OS and RS Dimeglio score.

## Data Availability

Not applicable.

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
