# Peer review of "Interobserver Reliability of Pirani and Dimeglio Scores in the Clinical Evaluation of Idiopathic Congenital Clubfoot"

_children, 2021, doi:10.3390/children8080618_

Round 1
Reviewer 1 Report
Dear authors,
Thank you very much for the possibility of reviewing this interesting manuscript on the interobserver reliability of Pirani and Dimeglio scores. The content of the paper is relevant to the readership and falls within the scope of the journal, and I enjoyed reading the manuscript.
However, I suggest to address the following minor methodological and orthographic issues:
Specific comments:
l. 15, l. 24f: The term “clinical setting” might be confusing – maybe better “clinical outcome”?
l. 21: “in fully trained paediatric orthopaedic surgeons”?
l. 23: What does the period of 6 months refer to – 6 months of experience for the orthopaedic surgeons with the scoring systems, or 6 months of observation of children with CTEV?
l. 50f: Maybe you could also state in which year the Pirani score was introduced.
l. 52: Redundant definition of the abbrevations DimS and PirS, which have already been introduced before (l. 46 and 50, respectively).
l. 62 “…, a review of all infants younger than…” ?
l. 68f: “The inclusion criteria were…: (2) bilateral CTEV” – were patients with unilateral CTEV really excluded? If so, could you explain why?
l. 83: “…where a higher score indicates…”
l. 100: “All the observers underwent 1 h of training” – what kind of training? Presumably not a training on the scoring systems, as you state that all evaluators already had previous experience of at least 6 months with the scoring systems?
l. 116: “…; between the 0.7 and 0.89 as “strong,..” – redundant “the”.
l. 124: “The median of cast numbers” – plural?
l. 162f: “In pediatric orthopaedic disorders assessed by experienced observers, greater concordance in the assessment.” – please consider revising this sentence, as it seems to be incomplete.
l. 176: “every sub-parameter highlighted poor reliability values expert for..:” – except for?
l. 189f: “Despite good reliability and a strong correlation between in the NPeO group” – redundant “between”?
l. 192: Maybe better: “It can be assumed the routine practice in clubfoot treatment can provide…”?
l. 199: “…in the clinical evaluation of CTEV,…”
Author Response
Dear authors,
Q) Thank you very much for the possibility of reviewing this interesting manuscript on the interobserver reliability of Pirani and Dimeglio scores. The content of the paper is relevant to the readership and falls within the scope of the journal, and I enjoyed reading the manuscript.
A) Thank for your constructive comments. After the revision we believed that the manuscript improved its quality.
However, I suggest to address the following minor methodological and orthographic issues:
Specific comments:
Q1) l. 15, l. 24f: The term “clinical setting” might be confusing – maybe better “clinical outcome”?
Q2) l. 21: “in fully trained paediatric orthopaedic surgeons”?
A1 and A2) Thank for your comments. The requested modifies were performed
Q3) l. 23: What does the period of 6 months refer to – 6 months of experience for the orthopaedic surgeons with the scoring systems, or 6 months of observation of children with CTEV?
A3) Additional information was included in the abstract.
Q4) l. 50f: Maybe you could also state in which year the Pirani score was introduced.
A4) Pirani Score first description year was included in the introduction.
Q5) l. 52: Redundant definition of the abbrevations DimS and PirS, which have already been introduced before (l. 46 and 50, respectively).
A5) The redundance was eliminated.
Q6) l. 62 “…, a review of all infants younger than…” ?
l. 83: “…where a higher score indicates…”
l. 116: “…; between the 0.7 and 0.89 as “strong,..” – redundant “the”.
l. 124: “The median of cast numbers” – plural?
l. 162f: “In pediatric orthopaedic disorders assessed by experienced observers, greater concordance in the assessment.” – please consider revising this sentence, as it seems to be incomplete.
l. 176: “every sub-parameter highlighted poor reliability values expert for..:” – except for?
l. 189f: “Despite good reliability and a strong correlation between in the NPeO group” – redundant “between”?
l. 192: Maybe better: “It can be assumed the routine practice in clubfoot treatment can provide…”?
l. 199: “…in the clinical evaluation of CTEV,…”
A6) Thank for your comments. The requested modifies were performed
Q7) l. 68f: “The inclusion criteria were…: (2) bilateral CTEV” – were patients with unilateral CTEV really excluded? If so, could you explain why?
A7) The unilateral clubfeet were excluded to avoid sample selection bias, similar clinical cases were assessed in order to avoid any external influence.
l. 100: “All the observers underwent 1 h of training” – what kind of training? Presumably not a training on the scoring systems, as you state that all evaluators already had previous experience of at least 6 months with the scoring systems?
A8) Despite at least 6 months of scoring systems experience, the observers underwent to theoretical CTEV clinical manifestation and scoring system training in order to avoid any assessment bias due to the low or not routinely clinical practice.
Reviewer 2 Report
This study assessed reliability of Pirani and Dimeglio scores in the evaluation of congenital clubfoot. A total of 6 physicians (3 orthopaedic surgeons and 3 resident doctors) evaluated 54 feet (27 patients). My main comment is the small size in this study, which might lead to chance findings. With a total of 6 physicians' evaluation, it is difficult for the study to conclude on the reliability of the two scores.
Author Response
Thank you for your comment. The strength of the study is the large observers’ sample, in fact, our study included 6 contributors for a total of 324 CTEV evaluation; Previously published articles include in the assessment between one and three observers with a range between 38 and 442 CTEV evaluations; Moreover, our article is the first to assess the Pirani and Dimeglio score Interobserver reliability between different experience level physicians. For these reasons, we believe that the study could be a novelty in the literature.
We will be proud and honored if you may re-evaluate this study.
Round 2
Reviewer 2 Report
- I might have missed this in the first review, but does Figure 1 indicate OS tend to score lower than RS when the scores are lower, and the opposite when the scores are higher? Is this something meaningful?
- A minor comment in the abstract conclusion: "However, more experience 24 with CTEV leads to a better clinical setting", what does "clinical setting" mean? is it similar to clinical evaluation or assessment?
- Please check a few grammar issues. E.G., in first paragraph of Discussion "greater concordance in the assessment was founded", is it "found"? Later in the Discussion, "In both the cohorts", is it " In both cohorts"?
Author Response
Thank you for your comments.
All checked typos have been corrected.
Figure 1 did not show any statistical significance.